# New Treatment Targets and Innovative Lipid-Lowering Therapies in Very-High-Risk Patients with Cardiovascular Disease

**DOI:** 10.3390/biomedicines10050970

**Published:** 2022-04-22

**Authors:** Achim Leo Burger, Edita Pogran, Marie Muthspiel, Christoph Clemens Kaufmann, Bernhard Jäger, Kurt Huber

**Affiliations:** 13rd Medical Department with Cardiology and Intensive Care Medicine, Clinic Ottakring (Wilhelminenhospital), Montleartstrasse 37, 1160 Vienna, Austria; achim.leo.burger@gmail.com (A.L.B.); edita.pogran@gmail.com (E.P.); marie.muthspiel@gmx.at (M.M.); christoph.c.kaufmann@gmail.com (C.C.K.); bernhard.jaeger@meduniwien.ac.at (B.J.); 2Medical School, Sigmund Freud University, 1020 Vienna, Austria

**Keywords:** lipid-lowering therapy, PCSK9, the lower the better, strike effective and strong, inclisiran, bempedoic acid, alirocumab, evolocumab

## Abstract

The effective and fast reduction of circulating low-density lipoprotein cholesterol (LDL-C) is a cornerstone for secondary prevention of atherosclerotic disease progression. Despite the substantial lipid-lowering effects of the established treatment option with statins and ezetimibe, a significant proportion of very-high-risk patients with cardiovascular disease do not reach the recommended treatment goal of <55 mg/dL (<1.4 mmol/L). Novel lipid-lowering agents, including the proprotein convertase subtilisin/kexin type 9 (PCSK9) antibodies alirocumab and evolocumab, the small interfering ribonucleotide acid (si-RNA) inclisiran, as well as the recently approved bempedoic acid, now complete the current arsenal of LDL-C lowering agents. These innovative therapies have demonstrated promising results in clinical studies. Besides a strong reduction of LDL-C by use of highly effective agents, there is still discussion as to whether a very rapid achievement of the treatment goal should be a new strategic approach in lipid-lowering therapy. In this review, we summarize evidence for the lipid-modifying properties of these novel agents and their safety profiles, and discuss their potential pleiotropic effects beyond LDL-C reduction (if any) as well as their effects on clinical endpoints as cardiovascular mortality. In addition to a treatment strategy of “the lower, the better”, we also discuss the concept of “the earlier, the better”, which may also add to the early clinical benefit of large LDL-C reduction after an acute ischemic event.

## 1. Introduction

Patients with established atherosclerotic cardiovascular disease (ASCVD) are a high-risk population for ischemic events [1,2]. Numerous studies have consistently demonstrated a significant reduction in ischemic events through lowering of low-density lipoprotein-cholesterol (LDL-C) [1,3,4]. Treatment with statins has been the cornerstone of lipid-lowering therapy in the past three decades, as several landmark trials have provided strong evidence of cardiovascular risk reduction by statins in secondary prevention [5,6,7,8]. Additional reduction of LDL-C and atherosclerotic events can be achieved by concomitant treatment with ezetimibe, a Niemann–Pick C1 inhibitor that reduces the intestinal absorption of cholesterol [9]. Nevertheless, LDL-C reduction below the recommended treatment target of 55 mg/dL (1.4 mmol/L), as suggested for very-high-risk patients in the current guidelines from the European Society of Cardiology (ESC), is difficult to achieve in many patients with statin and ezetimibe treatment alone [10]. Novel lipid-lowering therapies have been developed in recent years and provide substantial LDL-C reduction on top of an established treatment with high-intensity statin plus/minus ezetimibe [11,12,13,14]. These novel treatment options include the proprotein convertase subtilisin/kexin type 9 (PCSK9) antibodies alirocumab and evolocumab, the small interfering ribonucleotide acid (si-RNA) inclisiran, and the cholesterol biosynthesis inhibitor bempedoic acid [11,12,13,14] (Figure 1).

### Novel Aspects of This Review

In this manuscript, we provide a comprehensive review of the current evidence, safety profiles, and potential pleiotropic effects of these novel lipid-lowering agents. In addition, we underline the importance of a pro-active approach in lipid-lowering therapy with a thorough discussion of the novel treatment concepts of “the lower, the better” and “strike effective and strong”. Consistent evidence supports the strategy to maximize LDL-C reduction in very-high-risk patients to very low LDL-C levels (<25 mg/dL; 0.65 mmol/L) without increasing potential adverse effects. Furthermore, we summarize and discuss the emerging evidence of the benefits of very fast LDL-C reduction after an ischemic event.

## 2. Inhibition of PCSK9 for LDL-C Reduction

The circulating protein PCSK9 plays a central role in LDL-C regulation and the expression of LDL-C receptors on the surface of hepatocytes [15,16]. Plasmatic LDL-C is bound by LDL-C receptors on liver cells to form a complex and to internalize as endosomes to the cytoplasm [17]. The acidic pH of these endosomes causes dissociation of the complex with LDL-C degraded in lysosomes and the LDL-C receptor recycled to the cell surface for further use [18]. PCSK9 circulates in the plasma and binds to LDL-C receptors, leading to internalisation into the liver cells [16]. In contrast to the complex of LDL-C with the LDL-C receptor, the binding of PCSK9 to the LDL-C receptor is supported by enhanced affinity and does not dissociate at low pH levels [18]. The complex of the LDL-C receptor and PCSK9 is transferred to the lysosome for degradation and LDL-C receptor recycling on the cell surface of the liver cells is significantly reduced [19,20]. Consequently, LDL-C receptor expression and LDL-C uptake are significantly lowered [21]. The inhibition of PCSK9 increases the presentation of LDL-C receptors on the liver cells and is a highly effective approach to achieve substantial LDL-C reduction [21].

## 3. Monoclonal Antibodies for PCSK9 Inhibition—Alirocumab and Evolocumab

Alirocumab is a fully humanized monoclonal antibody with a maximal suppression of circulating PCSK9 within four to eight hours and a half-life during steady state of 17 to 20 days [22]. The treatment is administered subcutaneously as 150 mg every two weeks or 300 mg once a month and leads to a substantial reduction of LDL-C [22]. The lipid-lowering efficacy was confirmed in a meta-analysis of randomized trials investigating alirocumab on top of a maximally tolerated statin-therapy in patients at high cardiovascular risk [23]. This analysis demonstrated that treatment with alirocumab leads to a further LDL-C reduction of 60.5% (95%CI: 53.9–66.7%) when added to a baseline statin therapy [23]. By use of a lipid-lowering triple combination including high-intensity statin, ezetimibe, and alirocumab, an LDL-C reduction of 85% from treatment-naive baseline levels can be achieved [10,24]. In addition, the concentrations of other lipoproteins, including lipoprotein (a) [Lp(a)], apolipoprotein B, and triglycerides, are significantly reduced under treatment with this PCSK9 antibody [24,25,26].

The effect on cardiovascular outcomes was assessed in the ODYSSEY OUTCOMES study [11]. This multicenter randomized study tested the efficacy of alirocumab versus placebo on top of maximally tolerated statin therapy in 18,924 patients with acute coronary syndrome 1–12 months prior to study inclusion [11]. After a median follow-up duration of 2.8 years, treatment with alirocumab significantly reduced the occurrence of the primary composite endpoint of major adverse cardiovascular events (MACE) compared to placebo (HR = 0.85, 95%CI: 0.78–0.93, *p* < 0.001) [11]. The greatest risk reduction was observed in patients with baseline LDL-C levels of more than 100 mg/dL (2.6 mmol/L) (HR = 0.76, 95%CI: 0.65–0.87, *p* < 0.001) [11]. Pre-specified sub-analyses demonstrated a reduced risk of all-cause death (HR = 0.83, 95%CI: 0.71–0.97, *p* = 0.02), nonfatal cardiovascular events (HR = 0.87, 95%CI: 0.82–0.93, *p* < 0.001), and ischemic stroke (HR = 0.73, 95%CI: 0.57–0.93, *p* = 0.01) in patients treated with alirocumab, without increasing the risk of hemorrhagic stroke (HR = 0.83, 95%CI: 0.42–1.65, *p* = 0.59) [27,28].

Evolocumab is a human monoclonal antibody that achieves maximal suppression of PCSK9 four hours after subcutaneous injection of the recommended 140 mg every two weeks or 420 mg once a month [29]. The half-life of this agent ranges between 11–17 days [29]. In a similar fashion to alirocumab, treatment with evolocumab leads to a profound and long-lasting LDL-C reduction. Results from the prospective Mendel-2 and Gauss-2 studies demonstrated that evolocumab reduced LDL-C by 53–57% compared to placebo in patients with documented statin intolerance [30,31]. Significant reductions were also observed for Lp(a), apolipoprotein B, and triglycerides [30,31].

The FOURIER study investigated the potential benefit of evolocumab on clinical outcomes [12]. A total of 27 564 patients with ASCVD and LDL-C >70 mg/dL on high-intensity statin treatment were randomized to receive therapy with evolocumab (140 mg every two weeks or 420 mg once a month) or placebo [12]. After 48 weeks of follow-up, patients treated with evolocumab had a significantly lower risk of the primary endpoint of MACE compared to placebo (HR = 0.85, 95%CI: 0.79–0.92, *p* < 0.001) [12]. The risk of myocardial infarction (HR = 0.73, 95%CI: 0.65–0.82, *p* < 0.001) or stroke (HR = 0.79, 95%CI: 0.66–0.95, *p* = 0.01) was significantly lower in the evolocumab group [12]. LDL-C was reduced by further 59% on top of high-intensity statin therapy [12].

Treatment with alirocumab or evolocumab is generally well tolerated. The most common adverse events of PCSK9 antibodies are injection site reaction, which occurs in 2.5–5.9% of patients, with a mild clinical course in most cases [24,30,32,33]. Initial concerns of an adverse effect on neurocognitive function, potentially due to very low LDL-C levels, were not confirmed [34]. The EBBINGHAUS study—a prospective substudy of the FOURIER trial—analyzed the neurocognitive function of patients treated with evolocumab compared to placebo [34]. After the median follow-up of 19 months, no significant difference in neuropsychological testing was found [34]. This result was corroborated in a meta-analysis by Hirsch et al., which did not find an increased risk of neurocognitive adverse events during PCSK9 antibody treatment [35].

Overally, PCSK9 antibodies are potent and effective lipid-lowering medications with rarely occurring side effects. However, the significant costs of this therapy need to be addressed. With regards to the limited financial resources of healthcare systems, cost-effectiveness analyses are an important factor to consider. Initial studies reported the limited cost-effectiveness of PCSK9 antibodies at early market prices, although a more favorable result was demonstrated with increasing cardiovascular risk [36,37]. A more recent study showed that PCSCK9 antibodies are cost effective in very-high-risk patients for cardiovascular secondary prevention [38]. Data on the cost-effectiveness of PCSK9 inhibition with inclisiran are limited and warrant future research with updated market prices [39]. Of note, these financial analyses may differ based on the respective healthcare systems, changes in treatment prices, and the estimated cost-effective threshold [40]. Nevertheless, it is important to state that access to these highly effective lipid-lowering therapies must be a top-priority in very-high-risk patients in need of further LDL-C reduction.

## 4. Inclisiran

The double-stranded siRNA inclisiran is an alternative treatment option for PCSK9 inhibition [41]. After subcutaneous injection, inclisiran inhibits the translation of PCSK9 specifically in hepatocytes via sequence-specific binding to PCKS9 mRNA and activation of the RNA-induced silencing complex (RISC) [41,42]. The activated RISC cleaves the mRNA molecules of PCSK9, which are then degraded and unavailable for translation [42]. Thus, the protein synthesis of PCSK9 is substantially reduced during inclisiran treatment [41]. Data from the phase III clinical studies of the ORION-10 and ORION-11 studies demonstrated that plasma levels of PCSK9 decrease by up to 80% after treatment initiation with inclisiran [14]. Lower plasma levels of PCSK9 lead to increased LDL-C receptor recycling and a markedly increased uptake of LDL-C into liver cells [43]. Consequently, treatment with inclisiran leads to a substantial reduction in LDL-C [14]. The randomized ORION-10 and ORION-11 studies included a total of 3178 patients with established ASCVD or an equivalent risk on the maximally tolerated statin dose [14]. After the follow-up period of 510 days, LDL-C was significantly (*p* < 0.001) reduced by 52.3% and 49.9% in the ORION-10 and ORION-11 studies compared to placebo, respectively [14]. The randomized ORION-9 study tested inclisiran in a population of 482 patients with heterozygote familial hypercholesterinemia on maximally tolerated statin therapy with or without ezetimibe [44]. After 510 days of follow-up, LDL-C was significantly (*p* < 0.001) reduced by 39.7% (95%CI: 35.7–43.7%) in the inclisiran group compared to placebo [44].

Despite the promising results of substantial LDL-C lowering capabilities, data on cardiovascular clinical outcomes from an adequately powered studies are not yet available [45]. However, a meta-analysis of the ORION-9, -10, and -11 trials reported a significant reduction in MACE by 24% (HR = 0.76, 95%CI: 0.61–0.92, *p* = 0.001) compared to placebo [46]. The still ongoing ORION-4 trial (NCT03705234) will clarify the effect of inclisiran on cardiovascular outcomes [45]. This study will randomize a total of 15,000 patients with prior cardiovascular disease and high-intensity statin therapy to treatment with inclisiran or placebo [45]. The primary completion date is estimated for July 2026 [45].

So far, no comparison of efficacy between inclisiran and one of the PCSK9 antibodies is available. However, the ongoing ORION-3 study [47] is an active comparator extension trial of the phase-1 ORION-1 study [48] and investigates the LDL-C lowering capabilities, safety, and tolerability of inclisiran compared to evolocumab. This trial will include a total study population of 382 patients with ASCVD or an equivalent cardiovascular risk and will provide valuable insights of the potential differences of PCSK9 inhibition via a monoclonal antibody or via reduced translation and protein synthesis [47]. Although the primary completion date of this study was in December 2021, no results have yet been published [47].

Inclisiran is generally well tolerated and has a good safety profile [41]. The phase I study ORION-1 reported no serious adverse events related to inclisiran treatment [41]. In the large ORION-10 and ORION-11 studies, local injection site reactions were reported significantly more frequently during inclisiran therapy compared to the placebo groups, but were considered mild in most cases [14]. No significant side effects with respect to liver or kidney function or muscle-related side effects were observed [14]. In contrast to the PCSK9 antibodies alirocumab and evolocumab, treatment with inclisiran requires subcutaneous injections only twice a year, which might positively affect patient’s adherence to therapy.

## 5. Bempedoic Acid

Bempedoic acid is a once daily, orally administered pro-drug that is converted to its active metabolite in the liver cells by the enzyme very long-chain acyl-coenzyme A synthetase 1 (ACSVL1) [49]. Activated bempedoic acid reduces endogenous cholesterol synthesis by inhibiting the adenosine triphosphate citrate lyase (ATP citrate lyase) upstream to the inhibition of statin treatment [49]. Reduced intracellular cholesterol synthesis in the hepatocytes and subsequently upregulated LDL-C receptor expression with increased LDL-C plasma clearance leads to significant lipid-lowering properties of bempedoic acid [49]

The CLEAR study program of randomized, placebo-controlled trials tested the efficacy and safety of bempedoic acid in patients with high cardiovascular risk [13,50,51,52]. The CLEAR Harmony study randomized 2230 patients with ASCVD or heterozygote familial hypercholesterinemia and an LDL-C > 70 mg/dL despite maximally tolerated statin therapy with or without additional lipid-lowering treatment (ezetimibe or fibrate) to receive bempedoic acid or placebo [13]. After 12 weeks of follow-up, treatment with bempedoic acid reduced LDL-C by 16.5% (19.2 mg/dL, 0.5 mmol/L) from baseline [13]. Patients treated with bempedoic acid had a numerically lower risk of MACE (4.6 vs. 5.7%, *p* = 0.30), although this study was not powered for cardiovascular outcomes [13]. The CLEAR Wisdom study included a similar study population of 779 patients with ASCVD or heterozygote familial hypercholesterinemia on maximally tolerated lipid-lowering therapy [50]. After 12 weeks of therapy, patients with bempedoic acid had significantly lower LDL-C levels compared to placebo (−15.1% vs. 2.4%, *p* < 0.001) [50]. The CLEAR Serenity study tested the efficacy of bempedoic acid in 345 patients with statin intolerance and elevated LDL-C despite background non-statin therapy (most commonly ezetimibe or fish oil) [51]. Treatment with bempedoic acid significantly reduced LDL-C compared to placebo (−23.1% vs. −1.3%, *p* < 0.001) after 12 weeks of follow-up [51]. The lipid-lowering effect was less pronounced in the subgroup analysis of patients with T2DM, although no interaction was observed for this parameter in other CLEAR studies [13,50,51,52]. The CLEAR Tranquility study included 269 patients with statin intolerance and concomitant treatment with ezetimibe [52]. Bempedoic acid reduced LDL-C by 23.5% from baseline in this cohort [52]. Of note, although investigating the efficacy of bempedoic acid in statin-intolerant patients, the CLEAR Serenity and Tranquility study included 8.4% and 31% of patients with low-dose statin therapy, respectively [51,52]. In addition to the pronounced LDL-C reduction, the CLEAR studies also demonstrated a significant reduction in non-HDL and apolipoprotein B [13,50]. The potential benefit on cardiovascular outcome is not yet clarified and is currently under evaluation in the still-ongoing CLEAR Outcomes study. This trial will include a total of 14,014 patients with statin intolerance and established or high risk of cardiovascular disease and will clarify the potential impact on cardiovascular events [53]. Primary results are expected for December 2022 [53].

Treatment with bempedoic acid presents an overall favorable safety profile [13,50,51]. In contrast to statins, myopathy seems to be insignificant during bempedoic acid treatment [13,50]. This may be explained by the necessity of activation in the liver cells by the enzyme ACSVL1, which is not expressed in skeletal muscles [49]. However, increased levels of uric acid were found more frequently in patients treated with bempedoic acid in the CLEAR studies, although the overall incidence of gout was low (1–2%) [13,50,51].

## 6. Pleiotropic Effects beyond LDL-C Reduction

Chronic inflammation is regarded as one of the most important factors in atherosclerotic disease progression [54]. While the major effect of statin treatment is the substantial reduction of LDL-C, additional beneficiary properties beyond the mere lipid-lowering effect have been described [4,55]. These pleiotropic effects include anti-inflammatory and anti-oxidant properties [55]. Previous research in experimental and clinical studies demonstrated that statins significantly reduce plasma levels of several inflammatory mediators, including high-sensitivity C-reactive protein (hs-CRP), interleukin (IL)-1, IL-6, and tumor necrosis factor alpha (TNF-alpha) [56,57,58]. In addition, statin treatment decreases the activation of thrombocytes, inhibits leukocyte migration and endothelial adhesion, and increases the bioavailability of nitric oxide [55,56]. However, potential pleiotropic effects of the novel lipid-lowering agents have been less thoroughly investigated.

PCSK9 is primarily produced in the hepatocytes, but significant expression can also be found in the intestine, the mesenchymal cells of the kidney, in the endothelial and smooth muscle cells of the vascular wall and in macrophages [59,60]. Previous studies indicated that PCSK9 may have an impact on vascular inflammation and platelet function. PCSK9 correlates with white blood cell count, attenuates the pro-inflammatory effect of oxidized LDL on macrophages and increases monocyte migration to atherosclerotic plaques [61,62,63]. Treatment with alirocumab was associated with decreased atherosclerotic lesion size and monocyte recruitment in a mouse model [64]. This finding was also observed in the GLAGOV and the recent HUYGENS studies, demonstrating that treatment with evolocumab significantly reduces atherosclerotic atheroma volume and improves plaque stability via an increase in fibrous cap thickness [65,66]. In addition, Marques et al. showed that treatment with alirocumab reduces activation and chemotaxis of neutrophils and eosinophils [67]. On the other hand, a meta-analysis of randomized controlled trials did not find a significant reduction of hsCRP during treatment with alirocumab or evolocumab [68].

Although PCSK9 has a substantial impact in atherosclerotic disease progression and treatment with inclisiran was shown to significantly reduce LDL-C, the evidence on its potential effects on inflammation or thrombocyte function is limited so far [69]. In a pre-specified substudy of the ORION-1 trial, inclisiran therapy was not associated with a significant alteration in the pro-inflammatory cytokines IL-6 and TNF-alpha [69]. Counts of platelets, leucocytes, or monocytes were not influenced by inclisiran therapy [69]. In addition, plasma levels of hs-CRP did not differ between patients treated with inclisiran or with placebo in the ORION-10 and -11 trials [14].

The CLEAR studies of bempedoic acid demonstrated not only a substantial lipid-lowering effect, but also a significant reduction of hsCRP [13,51]. A meta-analysis showed that treatment with bempedoic acid reduces hsCRP by −27.0% (95%CI: −32.4–22.6%, *p* < 0.001), indicating a systemic anti-inflammatory effect [70]. In vitro studies showed that bempedoic acid decreases the inflammatory response of monocytes via activation of AMP-activated protein kinase (AMPK), which has been demonstrated to decrease IL-1, IL-6, IL-8, and TNF-alpha [71,72]. In addition, treatment with bempedoic acid reduces the expression of proinflammatory and profibrotic genes in liver cells in animal models, which was associated with a significant improvement as assessed by the non-alcoholic fatty liver disease score [73].

When discussing all these potential pleiotropic mechanisms, it is important to note that the anti-inflammatory or anti-atherosclerotic actions of all lipid-lowering agents can only be proven by use of extremely high doses of the agents. Accordingly, it remains unproven whether lipid-lowering agents induce such anti-atherosclerotic properties by direct action or by indirect actions via LDL-C reduction.

## 7. LDL-C Treatment Targets—The Lower the Better

Patients with established ASCVD greatly benefit from LDL-C reduction [10,74]. In addition to the treatment target of <55 mg/dL (1.4 mmol/L) in very-high-risk patients with documented ASCVD, a further reduction below 40 mg/dL (1.0 mmol/L) may be considered in patients with a second vascular event within two years [10]. Although no randomized controlled study investigated these treatment targets specifically, these recommendations are supported by sub-analyses of the FOURIER and the ODYSSEY OUTCOMES studies [75,76]. In the FOURIER trial, a total of 2669 patients reached very low levels of LDL-C (<20 mg/dL, 0.5 mmol/L) after four weeks of treatment with evolocumab [75]. These patients had the lowest risk for ischemic events in the whole study population, without an increase in adverse events [75]. Likewise, a propensity score-matched analysis of the ODYSSEY Outcomes study reported that patients achieving an LDL-C of <25 mg/dL (0.65 mmol/L) had a particularly low risk of MACE without an excess risk of hemorrhagic stroke or dementia [76]. In contrast to the J-shaped curve of hypertension treatment [77], the benefit of LDL-C reduction follows a monotonic relationship with the lowest risk for ischemic events in patients with the lowest LDL-C levels [75,76]. This dose-dependent association between cardiovascular events and LDL-C reduction was consistently demonstrated in numerous studies [78,79,80]. A meta-analysis of the “Cholesterol Treatment Trialists’ Collaboration” with data from 170,000 patients of 26 randomized trials demonstrated that a reduction of LDL-C per 1 mmol/L (39 mg/dL) is associated with a 10% (RR = 0.90, 95%CI: 0.87–0.93, *p* = 0.001) reduction in all-cause mortality and a 20% lower risk for cardiovascular mortality due to coronary heart disease (RR = 0.80, 95%CI: 0.74–0.89, *p* < 0.001) [81]. Patients with high baseline LDL-C levels and a large absolute LDL-C reduction benefit from a particularly great risk reduction, but also patients with lower baseline LDL-C levels significantly benefit from further lipid-lowering therapy [78,79,80,81].

Concerns have been raised with regard to potential harm and an increased risk of dementia in patients that achieve low levels of LDL-C cholesterol. However, these previous concerns were dismissed in several studies [82,83,84]. Patients with genetic variants that are associated with lifelong very low LDL-C levels have a considerable lower risk of cardiovascular disease without a significantly increased risk for other comorbidities or impaired cognitive function [85,86]. Patients with familial hypolipoproteinemia often have lifelong LDL-C levels of <50 mg/dL (1.3 mmol/L) due to a mutation in the apoB gene that leads to substantially decreased secretion of apoB-containing lipoproteins from liver cells [87]. Although an increased risk of hepatic steatosis was reported in these patients, no influence on cognitive function was documented [88]. The loss-of-function of PCSK9 is associated with low levels of LDL-C and a significantly lower risk for atherosclerotic events without an increase in other comorbidities [85,89]. A slightly increased risk for type 2 diabetes mellitus in this patient group was not confirmed in another study [85,90].

Despite the compelling benefit of LDL-C reduction and clear guideline recommendations, a significant proportion of patients do not meet required treatment targets in clinical practice. The recently published DA VINCI Study was an EU-wide observational study on LDL-C goal achievement and demonstrated that only 18% (95%CI: 17–20) of very-high-risk patients met the treatment target of LDL-C <55 mg/dL (1.4 mmol/L) [91]. Similar results were reported from the Euroaspire V survey [92], demonstrating that 71% of very-high-risk patients did not achieve an LDL-C reduction below the 2016 ESC treatment target of LDL-C <70 mg/dL (1.8 mmol/L) [93]. While treatment awareness might still play a significant role in this matter [91,94], the novel lipid-lowering agents, alone or as combination therapy with statins or ezetimibe, provide highly effective and well-tolerated treatment options to achieve sufficient LDL-C reduction below treatment targets [11,13,14,95].

## 8. Importance of Early Reduction below Treatment Target—“Strike Effective and Strong”

Current guidelines recommend a stepwise intensification of lipid-lowering therapy with treatment controls after 4–6 weeks [10]. Accordingly, in patients with high baseline LDL-C values that require an early start with a lipid-lowering triple therapy (Statin + ezetimibe + PCSK9 inhibition), the achievement of the recommended LDL-C goal can last up to 3 months [10]. This time delay until optimal lipid-lowering therapy might be particularly relevant as the risk of a recurrent myocardial infarction is highest in the early phase after the index event [96]. An early and strong LDL-C reduction in patients after an acute ischemic event might prevent recurrent thrombo-ischemic complications and could improve the long-term prognosis. Consequently, an innovative approach in lipid-lowering therapy has been proposed by the acute cardiovascular care society of ESC, to “strike effective and strong” (Figure 2).

This concept of an early and substantial LDL-C reduction after an ischemic event is supported by emerging evidence. Fonarow et al. demonstrated, in a large study cohort of more than 170,000 patients with acute myocardial infarction, that the initiation of statin therapy within the first 24 h after hospital admission significantly reduced overall mortality compared to patients without statin treatment (4.0 vs. 15.4%, *p* < 0.001) [97]. A meta-analysis of randomized controlled statin trials investigated the association between the timing of treatment initiation and the benefit on cardiovascular outcome [98]. This study showed that patients treated with statins even before percutaneous coronary intervention (PCI) benefit from the greatest risk reduction of recurrent myocardial infarction [98]. A recent nationwide Swedish cohort study analyzed the effect of early and strong LDL-C reduction in patients after myocardial infarction [99]. This study demonstrated that the patient quartile with the strongest absolute LDL-C reduction (1.85 mmol/L, 71 mg/dL) within 6–10 weeks after the index event had a significantly lower risk of MACE (HR = 0.77, *p* < 0.05) and recurrent myocardial infarction (HR = 0.81, *p* < 0.05) compared to the quartile with the least LDL-C reduction (0.36 mmol/L, 14 mg/dL) [99]. The randomized multicenter SECURE-PCI ACS trial investigated if prior loading with atorvastatin in patients with acute coronary syndrome and planned invasive management reduced the risk of MACE [100]. This study showed that the initiation of statin therapy prior to PCI was associated with a 28% relative risk reduction in MACE after 30 days of follow-up (*p* = 0.02) [100]. The treatment benefit was consistent in patients that received atorvastatin loading less than 12 h or less than two hours prior to PCI [101]. The randomized EVOPACS study assessed the efficacy of evolocumab versus placebo on top of high-intensity statin therapy for early LDL-C reduction in 308 patients with ACS [102]. Patients that were randomized to the evolocumab group had significantly (*p* < 0.001) lower levels of LDL-C after eight weeks of follow-up compared to the placebo group (0.79 mmol/L [31 mg/dL] vs. 2.06 mmol/L [80 mg/dL], *p* < 0.001) [102]. The treatment target of LDL-C < 55 mg/dL was achieved in 95.7% of patients in the evolocumab group compared to 37.6% in the control group [102]. Recurrent cardiovascular events were similar between the groups, although the study was not powered for this outcome parameter [102]. The randomized EVACS study investigated the effect of evolocumab versus placebo on top of high-intensity statin therapy in the early post-infarct period [103]. A total of 57 patients with non-ST elevation myocardial infarction (N-STEMI) were enrolled and randomized within 24 h after hospital admission [103]. Therapy with evolocumab was associated with a significant LDL-C reduction after 72 h of treatment compared to placebo (49 mg/dL [1.27 mmol/L] vs. 76 mg/dL [1.97 mmol/L], *p* = 0.02), demonstrating the feasibility of early LDL-C lowering with PCSK9 inhibition [103].

The currently ongoing, randomized AMUNDSEN study is the first trial to investigate the efficacy of evolocumab started before PCI in patients with STEMI or N-STEMI [104]. This multicenter study will include a total of 1666 patients randomized to receive either evolocumab before PCI or standard care [104]. The primary outcome parameter is the LDL-C reduction below 55 mg/dL after 12 months with overall mortality and ischemic events as secondary outcome parameters [104]. The primary completion date is expected to be in September 2023 [104].

## 9. Conclusions

Effective LDL-C reduction is of paramount importance for adequate cardiovascular secondary prevention. Novel lipid-lowering agents have shown favorable safety profiles and substantial LDL-C lowering capabilities. In combination with high-intensity statin and ezetimibe therapy, a relative LDL-C reduction of up to 85% from baseline values can be achieved. This is especially important regarding the monotonic relationship between LDL-C levels and the risk of ischemic events, supporting the approach of “the lower, the better”. In addition, emerging evidence has shown that early initiation of lipid-lowering therapy in ACS patients even before PCI is associated with favorable outcomes. In this very-high-risk population, we should “strike effective and strong”.

## Figures and Tables

**Figure 1 biomedicines-10-00970-f001:**
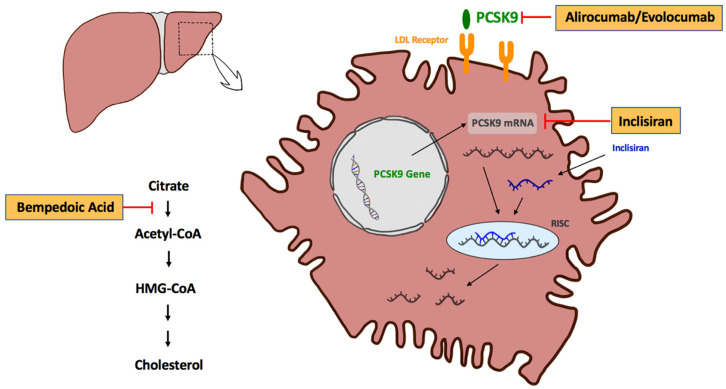
Mechanism of action of novel lipid-lowering agents. Acetyl-CoA: acetyl-coenzyme A, HMG-CoA: hydroxymethylglutaryl-coenzyme A, LDL: low-density lipoprotein, mRNA: messenger ribonucleotide acid, PCSK9: proprotein convertase subtilisin/kexin type 9, RISC: RNA-induced silencing complex.

**Figure 2 biomedicines-10-00970-f002:**
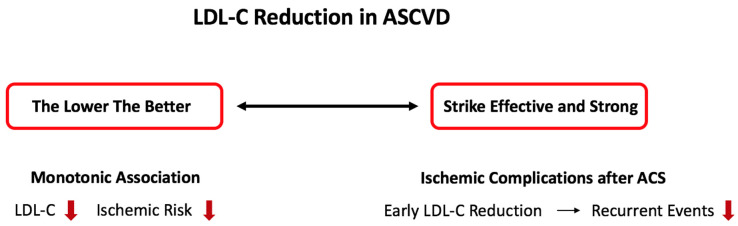
Treatment strategies in lipid-therapy: “the lower, the better” and “strike effective and strong”. ACS: acute coronary syndrome, ASCVD: atherosclerotic cardiovascular disease, LDL-C: low-density lipoprotein cholesterol.

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
