# Peer review of "New Treatment Targets and Innovative Lipid-Lowering Therapies in Very-High-Risk Patients with Cardiovascular Disease"

_biomedicines, 2022, doi:10.3390/biomedicines10050970_

Round 1

Reviewer 1 Report

This is a nice and logically edited overview of the currently available new treatment targets and innovative lipid-lowering therapies. The manuscript is well-written.

Comments:

  • There are dozens of highly similar reviews in the literature published in the recent few years. Therefore, the authors should highlight the new elements in a section entitled Novel aspects of this review
  • Although these novel therapeutic agents are widely available in many countries worldwide, because of the high costs, not all very high-risk patients receive PCSK9 inhibitors. The authors should discuss the financial difficulties and its consequences.
  • Despite the clear recommendations of the current guidelines, there are significant problems in the everyday clinical practice. Results of the Da Vinci study should be cited and discussed (PMID: 33580789).

Author Response

There are dozens of highly similar reviews in the literature published in the recent few years. Therefore, the authors should highlight the new elements in a section entitled Novel aspects of this review.

As suggested by the reviewer, we added the following paragraph to the manuscript, please see on page 2, lines 54-62:

Novel aspects of this review

In this manuscript, we provide a comprehensive review of the current evidence, safety profiles and potential pleiotropic effects of these novel lipid-lowering agents. In addition, we underline the importance of a pro-active approach in lipid-lowering therapy with a thorough discussion of the novel treatment concepts of “the lower – the better” and “strike effective and strong”. Consistent evidence supports the strategy to maximize LDL-C reduction in very high-risk patients to very low LDL-C levels (<25mg/dL; 0.65mmol/L) without increasing potential adverse effects. Furthermore, we summarize and discuss emerging evidence of the benefits of a very fast LDL-C reduction after an ischemic event.

Although these novel therapeutic agents are widely available in many countries worldwide, because of the high costs, not all very high-risk patients receive PCSK9 inhibitors. The authors should discuss the financial difficulties and its consequences.

 Following the suggestion of the reviewer, we added the following paragraph to the manuscript, please see on page 4, lines 133-145:

“Altogether, PCSK9 antibodies are potent and effective lipid-lowering medications with rare side effects. However, the significant costs of this therapy need to be addressed. In regards with the limited financial resources of healthcare systems, cost-effectiveness analyses are an important factor to consider. Initial studies reported a limited cost-effectiveness of PCSK9 antibodies at early market  prices, although a more favorable result was demonstrated with increasing cardiovascular risk (36, 37). A more recent study showed that PCSCK9 antibodies are cost effective in very high-risk patients in cardiovascular secondary prevention (38). Data of the cost-effectiveness of PCSK9 inhibition with inclisiran is limited and warrants future research with updated market prices (39). Of note, these financial analyses may differ based on the respective healthcare systems, changes in treatment prices and the estimated cost-effective threshold (40). Nevertheless, it is important to state that access to these highly effective lipid-lowering therapies must be a top-priority in very high-risk patients in need of further LDL-C reduction.”

Despite the clear recommendations of the current guidelines, there are significant problems in the everyday clinical practice. Results of the Da Vinci study should be cited and discussed (PMID: 33580789).

We thank the reviewer for this interesting and relevant comment. Accordingly, we added the following sentences to the manuscript. Please see the resulting changes on page 7, lines 326-337:

“Despite the compelling benefit of LDL-C reduction and clear guideline recom-mendations, a significant proportion of patients do not meet required treatment targets in clinical practice. The recently published DA VINCI Study was an EU-wide observational study on LDL-C goal achievement and demonstrated that only 18% (95%CI: 17-20) of very high-risk patients met the treatment target of LDL-C < 55mg/dL (1.4mmol/L) (91). Similar results were reported from the Euroaspire V survey (92), demonstrating that 71% of very high-risk patients did not achieve an LDL-C reduction below the 2016 ESC treatment target of LDL-C <70mg/dL (1.8mmol/L) (93). While treatment awareness might still play a significant role in this matter (91, 94), the novel lipid-lowering agents, alone or as combination therapy with statins or ezetimibe, provide highly effective and well-tolerated treatment options to achieve sufficient LDL-C reduction below treatment targets (11, 13, 14, 95).”    

Reviewer 2 Report

Burger et al. summarized the evidence of the lipid-modifying properties of the novel agents such as proprotein convertase subtilisin/kexin type 9 (PCSK9) antibodies alirocumab and evolocumab, the small interfering ribonucleotide acid (si-RNA) inclisiran and recently approved bempedoic acid. They review their safety profiles, potential pleiotropic effects and their effects on clinical endpoints as cardiovascular mortality. The manuscript is very well written and definitely deserves publication, however I have one major and one minor concern pointed out below:

Major concern

The ORION3 study (NCT03060577) should be included in this review. In my opinion this would be the only comparison between inclisiran and PCSK9 inhibitor (evolocumab). The results of this study will definitely provide the best comparison between these two drugs as I believe no other hard endpoint studies won’t be allowed by the sponsors. The ORION3 began in 2017 but still no results are available.

Minor

In Section 5 discussing Bempedoic acid the authors should clearly state which study was performed in lipid-lowering naïve patients and those already treated with statins.

Round 2

Reviewer 1 Report

Based on the modifications and the response of the authors, I believe the manuscript has been sufficiently improved and could be published in Biomedicines. 

Reviewer 2 Report

No further comments.